# "A good day is just being able to breathe": Aligning COPD research with patient needs, a qualitative study

Laurel O'Connor[1]*, Julia Ferranto[1], Anuska Ganesh Harne[1], Leah Dunkel[1], Peter Lindenauer[2], Bruce Miller[3], Christopher Mosher[4], Fernando Martinez[5], Apurv Soni[1]

1 Program in Digital Medicine, Department of Medicine, University of Massachusetts Chan Medical School, Worcester, Massachusetts, United States of America, 2 Department of Healthcare Delivery and Population Sciences and Department of Medicine, University of Massachusetts Chan Medical School–Baystate, Springfield, Massachusetts United States of America, 3 COPD Foundation, Miami, Florida United States of America, 4 Division of Pulmonary, Allergy, and Critical Care Medicine, Department of Medicine, Duke University School of Medicine, Durham, North Carolina, United States of America, 5 Division of Pulmonary, Allergy, and Critical Care Medicine, Department of Medicine, University of Massachusetts Chan Medical School, Worcester, Massachusetts, United States of America

* laurel.oconnor@umassmed.edu

## Abstract

### Background

Chronic obstructive pulmonary disease (COPD) is a common and impactful disease that is the target of a large portfolio of clinical research. However, there is limited understanding of how individuals with COPD perceive trial designs, outcomes, and intervention acceptability. The objective of this project was to explore the perspectives and priorities of patients and their caregivers toward COPD-focused clinical research.

### Methods

Semi-structured interviews were conducted with participants living with COPD and their caregivers using the Theoretical Framework of Acceptability (TFA) to guide data collection and analysis. Interviews were transcribed and coded using qualitative analysis software and analyzed using an inductive thematic approach.

### Results

Fifteen interviews were performed. Key themes included participant preference for outcome measures that directly impact daily living, such as mental wellness and physical function. Participants highlighted the need for research data to be actionable, advocating for health insights to be shared with participants and their healthcare providers. Study engagement was influenced by the perceived burden and

**Data availability statement:** All relevant data are within the paper and its Supporting Information files.

**Funding:** The project was supported by the National Center for Advancing Translational Sciences, National Institutes of Health, through Grants KL2TR001454 and UL1-TR001453, and the National Heart, Lung, and Blood Institute through grant 1K23HL174454-01A1. The funders had no role in study design, data collection and analysis, decision to publish, or preparation of the manuscript. The content is solely the responsibility of the authors and does not necessarily represent the official views of the NIH.

**Competing interests:** The authors have declared that no competing interests exist.

complexity of interventions as well as their direct relevance to patients. Patients favored research designs that minimize physical and logistical challenges. Lastly, participants desired greater involvement in the research design process.

## Conclusions

Aligning COPD research with patient priorities requires incorporating meaningful outcome measures, reducing participation burdens, and fostering ongoing engagement. Integrating patient-centered approaches in study design can enhance recruitment, adherence, and the real-world impact of COPD interventions.

## Introduction

Chronic obstructive pulmonary disease (COPD) significantly diminishes patients' quality of life [1]. It can persistently impact their ability to breathe comfortably, perform daily activities, and maintain social and occupational engagement [1–5]. Frequent COPD exacerbations contribute to psychological and physical strain, increasing the risk of anxiety and depression while also resulting in emergency department visits, hospitalizations, and worsening disability [3,6,7]. The cumulative impact of COPD on mobility, independence, and overall well-being makes it one of the most burdensome chronic diseases globally [3,7,8]. Given its high morbidity and associated substantial healthcare costs, high-quality research is essential to developing interventions that reduce hospitalizations, enhance symptom control, and improve patients' quality of life. Investigating innovative treatment strategies, such as digital monitoring, personalized therapy, and mobile treatment platforms, can help improve disease management and quality of life if they are adopted by clinicians and patients [9–12].

While common outcomes measured in COPD research assess disease progression, treatment effectiveness, and patient well-being, symptom burden and patient-reported outcomes (PROs) are increasingly recognized as critical measures, capturing aspects such as dyspnea severity (e.g., the Modified Medical Research Council (mMRC) scale), chronic cough, sputum production, and fatigue [13,14]. Health-related quality of life is measured by validated tools like the COPD Assessment Test (CAT), which evaluates how the disease affects daily life, mobility, and emotional well-being [13,15]. Additionally, exercise tolerance and physical endurance are measured using tests like the six-minute walk test (6MWT) [16]. Emerging digital health tools and remote patient monitoring are also enabling the collection of biometric and behavioral data, such as oxygen saturation trends, activity levels, and medication adherence [9,12].

Prior studies evaluating patient perceptions of outcomes in COPD management show that breathlessness, fatigue, and the ability to engage in daily activities are prioritized over traditional physiological markers such as lung function [2,17]. However, there remains a limited understanding of what matters most to patients in the design and measurement of COPD research [6]. Despite the growing emphasis on patient-centered investigation, little is known about how individuals with COPD perceive study designs, measurement strategies, and intervention acceptability. The gap

in knowledge regarding patient priorities in research design highlights the need for further investigation into how COPD studies can better align with patients' preferences, ensuring the research is fulfilling, the results are meaningful, and generalizable to persons living with COPD.

Given the daily burden on patients living with COPD, determining the selection of meaningful outcomes in research is critical to improving care. Patients are more inclined to adopt new treatments, technologies, or care models when they see clear benefits aligned with their health priorities [18,19]. For example, if research focuses solely on improving lung function but neglects to measure improvements in breathlessness— a symptom that significantly impacts daily living—patients may be less likely to embrace the intervention, even if it is proven to be clinically effective [2,6,20]. If an intervention is designed with patient input, it is more likely to be acceptable, feasible, and usable in daily life [13,18]. Including patients in research design also helps ensure successful clinical trials; interventions designed with patient perspectives in mind are more likely to successfully meet recruitment goals, retain participants, and yield high-quality data [19]. Patients bring lived experiences to research design that highlight challenges and priorities investigators might overlook. The objective of this study is to describe patient priorities in the design and execution of clinical trials evaluating novel interventions for COPD treatment and care delivery.

## Methods

### Study design and setting

This study utilized qualitative methods to obtain and analyze data from stakeholders through semi-structured interviews. The theoretical framework of acceptability (TFA) was used to organize the study's interview guide and top-level codebook [21]. TFA consists of seven component constructs: affective attitude, burden, perceived effectiveness, ethicality, intervention coherence, opportunity costs, and self-efficacy, and is used to evaluate partners' perspectives on interventions before, during, and after participation [21].

The study staff was based at an urban academic tertiary care medical center in the northeastern United States that serves urban, suburban, and rural communities. The institution has numerous active and recently completed clinical trials of interventions related to COPD treatment and care delivery. This study complied with the consolidated criteria for best practices in reporting qualitative research [22] and was approved by the Institutional Review Board of the affiliate medical school (IRB Docket#: STUDY00000126).

### Selection of participants

Sampling was performed regionally, purposefully recruiting patients with a diagnosis of COPD and their caregivers. Some had recently participated in a clinical trial with an intervention focused on COPD care and treatment; others had not participated in a clinical trial. All participants were recruited via email. Participants were offered a $50 gift card for their participation. The study team intentionally balanced new invitations to sample a balanced demographic distribution of participants by age, sex, race, and ethnicity. Recruitment ceased when thematic saturation was reached. Participants had no preexisting professional relationship with the staff conducting the interviews. All participants were provided with a written Fact Sheet both at the time of recruitment and again immediately before their interview. They provided fully informed verbal consent to a member of the study team. A second member of the study team witnessed the consent before the start of the interview, and the consent was notated at the beginning of each participant's interview notes. The consent process and participant-facing Fact Sheet were approved under the authorizing IRB and complied with all relevant ethical regulations, including the Declaration of Helsinki. Recruitment began on October 1, 2024, and was completed on March 15, 2025.

### Data collection

The interview guide was organized by the TFA domains and informed by recent literature about COPD outcomes [13,17,21]. The interview guide was piloted with 2 "test participants" and revised before use with actual participants.

Interviews began with general questions about participants' experiences with their COPD and any prior clinical trial experience. Subsequently, participants were asked more specific questions about their experiences and perspectives towards different parts of clinical study design, including recruitment, the intervention, data collection instruments, measures, outcomes, compensation, and offboarding. The context of the participants' COPD diagnosis was emphasized (e.g., outcomes of importance in the perception of their COPD severity).

The content and focus of each interview varied with the interest of the participant. The guide contained both open- and closed-ended questions with additional probes to be used at the discretion of the interviewer. At regular intervals during the project, the study team met and collaboratively modified the interview guide to incorporate emerging insights and recurring themes that participants raised, and to add or remove questions and probes based on how much useful information they elicited. The final version of the guide is included in S1 File.

The study team conducted private, 60-minute, semi-structured interviews with participants via video-teleconferencing between October 2024 and March 2025. Interviews were performed by one physician, one research scientist, and two research assistants experienced in qualitative methods and structured interviewing techniques. Interviews were audio recorded and transcribed using Otter.ai artificial intelligence software (Mountain View, California). Subsequently, the transcript was reviewed by a member of the research team for accuracy, and all identifying information was removed before analysis.

## Analysis

Demographic data were summarized with descriptive statistics. Interview transcripts were imported into Atlas.ti (23.2.0, Berlin, Germany) qualitative analysis software for coding. The study team developed an initial top-level codebook based on prior literature and organized by TFA domains. Two team members conducted a line-by-line review of five randomly selected transcripts, applying codes deductively based on the initial codebook. Simultaneously, the study staff used an iterative, constant-comparative approach to identify recurring patterns and inductively capture emerging themes, which were then incorporated into the refined codebook. Krippendorff's alpha (a) was calculated after the first five transcripts were coded and found to be 0.86, demonstrating valid reliability between coders [23].

Transcripts were coded by five members of the research team, including a physician and a research scientist with expertise in qualitative analysis, as well as three research assistants formally trained in qualitative coding and analysis. Each transcript was coded independently by two team members, after which all coded transcripts were collectively reviewed. Discussions continued until a full consensus was reached on the finalized codes. Following each consensus meeting, transcripts were re-coded using the revised codebook to incorporate refinements. This process continued until no new themes emerged, indicating thematic saturation.

Once coding was complete, the themes from qualitative interviews were organized using the Fogg Behavior Model (FBM) to describe how various factors influenced patient behaviors [24]. The FBM is a psychological framework that explains behavior as a function of three key elements: motivation, ability, and prompts [24]. According to the model, behavior occurs when an individual has sufficient motivation, the ability to act, and an external or internal prompt that triggers it. Interview themes were categorized based on these elements to visualize how barriers and facilitators influence patient engagement with COPD-focused clinical trials. This approach provided a structured way to identify potential leverage points for improving patient engagement while designing COPD-focused research interventions.

## Patient personas

To contextualize patient perspectives and better understand how individual experiences shape attitudes toward COPD research, we developed individual profiles for each participant [25,26]. These profiles, intended for the study team to use to guide future patient-centered study design, were constructed using demographic information, clinical history, disease severity, healthcare utilization patterns, and qualitative insights from interviews. By integrating these elements, we aimed

to capture the diverse backgrounds and lived experiences that influence participants' views on clinical trial design, outcome measures, and participation barriers.

## Results

The study team approached 41 prospective participants and completed 15 interviews (37.5% response rate). Participant demographics are summarized in Table 1. Analysis of participant interviews identified several key themes related to their experiences, perceptions, and priorities toward research participation, outcome relevance, and engagement. In total, thirteen patients and two caregivers were recruited. One caregiver was a patient's spouse, and the other was a patient's child; both are the primary, live-in caretakers of a patient with COPD.

### Theme 1: Patients prioritize measures and outcomes that are directly relevant to their daily lives and lived experiences

**Patients were focused on their physical capacity to participate in day-to-day life.** Patients emphasized the importance of research measuring outcomes that directly impact their day-to-day lives, including the physical capacity to complete tasks of daily living and promote their overall sense of normalcy. Many expressed frustration with clinical measures that failed to reflect their lived experience (e.g., pulmonary function tests) or measured things in aggregate, long-term metrics (such as hospitalizations over a year). Patients consistently valued practical markers of physical function as a high-priority outcome measure, such as the ability to move a certain distance or complete a specific task. A patient noted: "It's…being able to go in the shower…that is a good day. I can't ambulate much without shortness of breath. So, a good day for me is just being able to do the things that I like to do…a good day for me is just not being sick, being able to breathe…Every day is different with COPD" (Participant 11, Female patient). Patients also noted that measures and outcomes should be communicated in clear, accessible language to improve understanding and reduce anxiety during their participation.

**Evaluating mental health metrics is paramount.** Patients and caregivers frequently identified mental health and well-being as being of great importance when considering the impact of their COPD diagnosis. They noted that the severity of their COPD-related symptoms and the perceived burden on their lives and those of their families were central

**Table 1. Participant demographics (N = 15).**

| Age | |
|---|---|
| Mean (Std Dev) | 69.8 (7.1) |
| Median (IQR) | 72 (8) |
| Range | 45, 84 |
| **Gender (n, %)** | |
| Male | 6 (40.0) |
| Female | 9 (60.0) |
| **Race (n, %)** | |
| America Indian | 0 (0) |
| Asian | 0 (0) |
| Black or Africa American | 3 (20.0) |
| Native Hawaiian/Pacific Islander | 0 (0) |
| White | 12 (80.0) |
| **Ethnicity (n, %)** | |
| Hispanic/Latino | 2 (13.3) |
| Non-Hispanic/Latino | 13 (86.7) |

to their overall sense of well-being. They noted that they preferred measures of overall mental health rather than surveys specific to COPD-related distress. Patients indicated that COPD-specific screeners may fail to capture certain ways in which the disease impacts them. For example, patients expressed concern about how their families and caregivers are burdened with their disease, as well as fear around the unknown regarding their future quality and quantity of life, which are not captured by many commonly-used COPD-focused surveys. One participant stated, "It's not just how I feel…it's how I feel about how I feel" (Participant 12, Male patient).

Patients also noted that emotional distress could influence overall adherence to pulmonary rehabilitation or other investigative interventions. One participant stated, "You know, how does stress fit into all of this? It isn't just about how you feel because you are having trouble breathing, but what's going on with the whole-body experience, which includes the mind. So how do you motivate somebody that's having to do all these exercises and has other things on their mind?" (Participant 9, Male patient). Participants felt that supporting, monitoring, and measuring mental health may help improve the success of interventions and allow investigators to contextualize other data, including study adherence and retention.

**Metrics of caregiver burden should be included in research.** The role of caregivers was highlighted as patients emphasized that their condition placed a burden on family members, reinforcing the importance of research addressing both patient and caregiver needs. Concern about caregiver burden also had a direct impact on patient well-being. Patients perceived that caregivers and family members were not often engaged when measuring the impact of an intervention, despite them being very involved in managing patients' COPD. One caregiver noted, "I am anywhere from his part-time to full-time caregiver, depending on the state he's in. So, we have some good days, and then we have not good days" (Participant 15, Female caregiver). Participants noted they would prefer that caregivers be directly evaluated both in how much an intervention burdens them (e.g., driving patients to appointments) and how its impact changes their overall perception of how their loved one's COPD symptoms impact them.

**Patients want study data to cross over to their clinical providers expeditiously.** There was a strong expectation that any real-time data collected in a study should be shared with healthcare providers. Many participants expressed frustration that data being collected for research, particularly if potentially clinically meaningful, was not made available to the clinical team in a timely way, if at all. They emphasized that integrating research data into routine healthcare visits immediately could improve patient-provider communication and offer early interventions for worsening symptoms, which appealed to them as an incentive for participating in the study, particularly given the limited access they have to their physicians. Patients felt that this was particularly important in the context of COPD because of the speed with which exacerbation symptoms can start and worsen. Example quotations from all subthemes are summarized in Table 2.

### Theme 2: Study designs that optimize engagement from patients with COPD will determine data quality

**The burden on participating patients should be considered in the intervention design.** Participants highlighted several factors that influenced their willingness to participate and remain engaged in studies. The physical burden of participation was a key concern, with most perceiving that virtual participation made research more accessible. Because COPD limits physical activity, patients expressed reluctance towards participating in interventions that required travel outside of their existing clinical appointments. Participants also discussed that interventions that allowed them to participate from home were easier to adhere to and often more comfortable because they had access to their home environment and didn't worry about sensitivity to the environment: one participant stated, "You have maybe a good, maybe one to two months…in New England that a COPD patient can go outside, because when it starts getting too hot and humid, they can't breathe. So, it's, it's, it's hard because of the weather as well, because do you have the heat, the cold, the viruses…They're scared. They don't want to leave the house. They're afraid to get sick" (Participant 13, Female caregiver).

In addition to the physical burden, patients frequently expressed that cognitive barriers, such as having to use sophisticated technology or multiple devices to participate in studies, dampened their enthusiasm to participate, and they

**Table 2. Theme 1: Patients prioritize measures and outcomes that are directly relevant to their daily lives and lived experiences.**

| Subthemes | Exemplar Quotes |
|---|---|
| Patients were focused on their physical capacity to participate in day-to-day life | I can usually tell when I get up in the morning…How my day is going to go, because my breathing, I know, in the summer mornings, I get up and, you know, I first thing I do is I, after I go to the bathroom, is I do my nebulizer because I but I can tell just when I get up and I go to the bathroom, if I'm breathing really hard, I know it's, you know, I'm staying in today. I'm not going anywhere. I'm not doing anything. You know, I can tell that even before I even after I have my nebulizer, it's still hard, but you know, there are other days I get up and it's like, I go to the bathroom. I say, Okay, I'm going to do my nebulizer now. And it's kind of like, oh, this is going to be a good day. My breathing…I can actually breathe without it. (Participant 8, Female patient)<br> My goal this spring is to get him outside and try to get him up to a quarter to a half a mile. I don't know if I'm going to be there. We have to do this respiratory endurance. Otherwise, he's going to feel like he can't keep up. When he has this shortness of breath. He shuts down. And he kind of has shut down to some degree now, and I'm trying to move him back towards some better place. I don't know if I'm going to succeed or not, but that's the goal. So, the exercise piece is super important here(Participant 15, Female caregiver) |
| Evaluating Mental health metrics is paramount | I had a enjoyed the contact with a person that I had and sometimes would share what was on my mind above and beyond the COPD exercise program, right?...We all have our own individual issues, and we all deal with them for better or for worse in our own way. And so, it's but it has an effect on your physical health. It has an effect if you feel like crap, mentally or emotionally, are you really going to do the exercises? (Participant 9, Male patient)<br>I think there's just a huge disconnect between the mental health side of COPD and the actual physical symptoms of what's happening to the patient. That's…basically, like, I just have, like, a strong opinion on that. I think there really needs to be and hopefully, like with this research study, more mental health providers. (Participant 13, Female Caregiver) |
| Metrics of caregiver burden should be included in research | It's also not…just me living with this disease—my wife has to deal with COPD too even those she isn't diagnosed. How much I can help her around the house, and when I have a bad day, it affects her life and her…overall sense of wellbeing. If you only looks at my symptoms and not the stress it puts on her, it's missing a part of the picture. (Participant 12, Male patient)<br>The burden...is my well-being and anxiety about my mother's status more so than like, 'how am I going to arrange this logistically?' So, it was much more like a general sense of well-being, rather than like a logistical burden that I'm talking about.(Participant 13, Female Caregiver)<br>I even woke up in a startled the other night because I was like…I think some of it is PTSD... if you're through those events [exacerbations] with them, you will ultimately become far more, you know, sensitive to it. And so…if you're tired as a caregiver, you may be more susceptible to being, like, kind of nervous all of the time. You become more nervous, and you got to…kind of calm yourself. (Participant 15, Female Caregiver) |
| Patients want study data to cross over to their clinical providers expeditiously | And so, you know that would also be comforting. For example, you know, if you if there was something that were coming through a cardiac nature, you know, be it changes in pulse rates and things like that, you know, they My, my, my specialist might be, might find that information useful and helpful, so I don't know, maybe I'm taking it one step too far here. But, yeah, any anything that is potentially actionable or whatnot that's picked up in a study such as this, I think should be shared, uh, within my case, it would be, you know [my primary care doctor]. (Participant 1, Male Patient)<br>It would also provide another piece of motivation, I think, for the participants to know that their physicians, are also benefiting from the real time data. You know, we see these people every six months, or we see them once a year. And, you know, something going into my file on a regular basis from a study. I think it would be useful for the patient and…the doctor. (Participant 3, Male Patient) |

appreciate simple interfaces with fewer onboarding steps in addition to robust support from research staff to navigate study procedures.

**Long-term relationships enhance engagement.** Participants universally identified sustained relationships with research personnel as a major factor in establishing and maintaining engagement in studies. Ongoing connection was perceived as a sign of genuine investment not just in data collection but also in participants' well-being, therefore reinforcing their willingness to engage in research.

**Patients want to be empowered to understand their data and use it to make decisions.** Many participants expressed a desire for immediate feedback on their health status during the study, particularly regarding symptom fluctuations and potential exacerbations. In studies where clinical data is being collected frequently, participants valued having results and their clinical meaning shared with them longitudinally so that they could understand the potential implications on their health. Additionally, patients voiced disappointment when they did not receive summaries of their individual progress or study results, stressing the importance of ensuring that research participation is a two-way exchange of information. When asked why being empowered to see and understand their data from studies was important to them, participants shared that they wanted to incorporate meaningful findings into any important COPD-related healthcare decisions. Participants also wanted to alleviate anxiety about whether the findings portended worsening illness. Example quotations from all subthemes are summarized in Table 3.

### Theme 3: Patients want to break down the "fourth wall" of research to be involved as partners in the research process

**Patients with COPD care deeply about advancing knowledge and treatment of their disease.** A unifying theme voiced by nearly every participant was a desire to be a part of improving how the scientific and healthcare communities evaluate and treat patients living with COPD. Participants noted that they often feel limited in their role in society because of the physical limitations posed by their COPD. However, they felt empowered by being uniquely positioned to be able to help advance the field of COPD research. They expressed optimism that participating in clinical research studies would directly improve their care and the care of others living with the disease.

**Table 3. Theme 2: Study designs that optimize engagement from patients with COPD will determine data quality.**

| Subthemes | Exemplar Quotes |
|---|---|
| The burden on participating patients should be considered in the intervention design | It was helpful, yeah, all from the comfort of your house, yeah? Plus, I can listen to my own music when I'm doing it. That is very important. Not everybody likes blue tangle. You heard of blue tangle?...It's a German vampire, goth band. It's not everybody's taste… but if I'm at home I am comfortable and I can listen to my own music in my own living room. (Participant 4, male patient)<br>And so, you got to think about people the age of the people that have COPD currently aren't very tech savvy, so that's where it became hard. Was to set up how, how to get on to your pulmonary rehab access that would, that was the main struggle versus, you know, bringing the patient into the facility or whatnot. I think that that was the hardest part, was the technical side of it, but that that was pretty much, that was the only drawback from it. (Participant 13, female caregiver). |
| Long-Term Relationships Enhance Engagement | And she did call me because she saw that I was having, you know, that I wasn't feeling well, and she called to see how I was doing, which I thought was awesome that…she's still called to check on me. (Participant 8, female patient).<br>And the coaching was very positive, making me stay on track, just keeping me motivated to stay in the study. So that was probably the most positive part of the study. (Participant 10, female patient) |
| Patients want to be empowered to understand their data and use it to make decisions | It would be helpful to have more coaching on flare ups like, what…to watch on the Fitbit? Okay, I never know if it's like a flare up or just a bad day…I really wasn't aware what to expect during a flare up. So, I think maybe some education on that would be helpful. (Participant 2, male patient)<br>Of I think, like when you get test results on the Mychart thing, or even with the [study app] it should be in in a language and written such that a person can understand it of Oh, yeah, your blood level was blah, blah, blah, for example, and that, that means you're doing okay, and you've got nothing to worry about. Or, you know, are you, you know, you had a lot of red blood cells, and that's a bad thing… But if it could be easier for a person to understand and not written in, you know, written in such a way that you don't need, you know, two or three college degrees to understand what the heck they're trying to what they're saying. So it should be in simple English, of you know, so a person can understand it. (Participant 5, female patient)<br>And the most disappointing thing was not getting the results of the study at the conclusion and also any progress that I was making, or just exactly what those readings actually meant. (Participant 10, female patient) |

**Patients feel more deeply connected to healthcare through study participation.** For many participants, research participation deepened their sense of connection to the healthcare system and other patients. Some noted that engaging with a structured study helped them feel more supported, informed, and empowered to manage their disease. The ability to interact with healthcare providers, researchers, and patient communities reinforced their engagement in their care, underscoring the potential for research to serve as an extension of healthcare delivery rather than an isolated experience. Several participants noted that the transition out of their clinical trial activities, when the study ended, was particularly challenging, citing that they felt like they were removed from something longitudinal, meaningful, and useful in providing clinical resources and support. Example quotations from all subthemes are summarized in Table 4.

## Fogg behavioral model

There was substantial variation in participant readiness for engagement and the underlying reasons. Some participants reported both high motivation and high ability, indicating strong readiness to participate in research with minimal support. Others expressed high motivation but faced significant barriers such as health limitations or logistical challenges. A subset of participants showed low motivation despite few practical barriers, while a smaller group exhibited both low motivation and low ability, suggesting the need for more intensive outreach and support.

## Participant profiles

Participant profiles are depicted in S1 Fig. Across participants, several common themes emerged, including the importance of increased self-efficacy in disease management through research participation, a desire for the pragmatic use of personal health data, and a need for ongoing engagement and support during and after the study. Many participants attributed their improved health awareness and early detection of complications to participation in studies. However, technical difficulties, lack of clear instructions, and challenges with study follow-up were frequent barriers to engagement. Another notable theme was in the transition out of the study, with many participants reporting a sense of loss or

Table 4. Theme 3: Patients want to break down the "fourth wall" of research to be involved as partners in the research process.

| Subtheme | Exemplar Quotes |
| --- | --- |
| Patients with COPD care deeply about advancing knowledge and treatment of their disease | I think it's, I think it's important for the participants to see that data, to know that their data matters, you know, and that it is going into a useful set of people that that hopefully can do something with it… My interest would be in giving the medical community as much information as I possibly could. It can only help me or my heirs. (Participant 2, male patient)<br>I find that if I can share what's going on with me, which is, you know, and I know everybody's different, but if you know, if you learn something from other people that have the disease…not just someone who, you know, knows about how it's supposed to what's how people are supposed to feel. You actually know how you feel, and they know how you feel, because everyone's going through the same thing. So, and I think it's important that you know people, or doctors or nurses have the information from someone who has [COPD] in order to treat people that have it. I you know, I was happy to do the study. (Participant 8, female patient) |
| Patients feel more deeply connected to healthcare through study participation | I've had a lot of care, attention, surgery…. So, I've always been connected to a point, but I think this study actually made me feel more connected, and that's comforting…It's just, it's made me think a lot more about my health. You know, just being connected to my health team and my data. (Participant 1, Male Patient)<br>The great part was that I made a lot of good friends, and we've started to have in our own group, and we set up our own Zoom meeting, and we talk and discuss stuff, and it's it. There was a lot of fabulous, fabulous stuff that came of it, and I'm extremely grateful that I was allowed to participate in it. And, you know, I've done a lot of wonderful things in information. And you know, it's tough living with a chronic illness, and then when you have other stuff added into it, other health problems, it makes it can make life difficult. But the important part is to always try to focus on the good stuff in life…and hold on to that. (Participant 7, Female Patient) |

disconnection when structured support ended. Many reported that the study provided a sense of community, improved symptom management strategies, and increased their awareness of COPD self-care techniques.

One major variance was in participant engagement with digital health interventions. While some participants were highly motivated and tech-savvy and reported easily integrating wearable devices and digital coaching into their routines, others found technology burdensome or difficult to use. Some struggled with understanding medical data, while others expressed frustration with the lack of real-time feedback or clear integration with their ongoing care. Additionally, perspectives on virtual versus in-person support varied. For example, some participants valued the convenience of virtual pulmonary rehabilitation, while others felt that in-person sessions were more effective for maintaining motivation and engagement.

Caregivers also provided distinct perspectives, highlighting the emotional and logistical burdens of managing COPD care for a loved one. Many expressed a need for better caregiver support and integration into research and healthcare planning.

## Discussion

COPD is a debilitating and common disease; advancements in prevention, treatment, and care delivery are needed to combat its morbidity. Engaging with participants living with COPD and understanding their priorities will help ensure that studies are more relevant, patient-centered, and yield higher-quality data [19,27]. The present study explored the perspectives of individuals living with COPD and those assuming the caregiver role regarding research design, outcome measurement, and study engagement. Our findings underscore the importance of ensuring that studies measure outcomes that meaningfully reflect patients' lived experiences, implement interventions that minimize participation burden while maximizing perceived benefit and clinical applicability, address patients' physical and psychosocial needs, and include them as meaningful scientific partners.

A central finding in this study was that patients prioritize outcome measures with tangible relevance to their daily lives. While traditional COPD research often emphasizes clinical metrics such as lung function, participants expressed a strong preference for measures that capture symptom burden, emotional well-being, and caregiver impact. These findings reflect previous survey-based and qualitative studies of patients living with COPD, which indicated that relieving breathlessness and increasing exercise capacity, as well as mitigating patient anxiety and depression, were of primary importance [28–31]. The findings also reflect the previous literature that emphasized patients' focus on the performance of daily physical tasks, such as activities of daily living or the ability to engage in social events [30,31].

Many participants described how their ability to perform daily activities and maintain independence dictated their quality of life. However, unlike the narrower COPD-specific measures such as CAT, investigators should consider more holistic instruments that capture overall mental health and quality of life. This finding expands on previous investigations by clarifying that mental health should be evaluated independently of COPD-related distress while reflecting the importance of previously described COPD-related anxiety and depression [6,30,31]. Additionally, measuring "indirect" outcome measures such as caregiver burden may provide a more holistic insight into the broader effectiveness of interventions. While previous studies have shown that more severe COPD is associated with higher caregiver burden, and that patients with COPD describe significant concerns about how their diagnosis will impact their close relationships, less is known about how interventions targeted toward COPD management impact caregiver burden [30–32].

Potential barriers to the implementation of these patient-prioritized measures, including both physical function relevant to their daily lives and more holistic assessments of mental health, include a lack of standardization or regulatory validation in COPD research, making them difficult to compare across studies or include in clinical trials with strict reporting requirements. Patient-defined functional goals and broad mental health constructs often introduce variability that reduces statistical power and complicates interpretation. In addition, these measures can be burdensome to administer and may require additional resources or infrastructure. As a result, researchers frequently default to more established, objective

endpoints that may not fully reflect patients' lived experiences. It may, however, be possible to adapt and validate existing instruments to address this gap.

Another key theme was the belief that data collected during clinical trials should be actionable and integrated into routine care concurrently, if possible. Participants expressed frustration that study-generated data, even when potentially clinically relevant, was not always shared with them or their healthcare providers. They viewed real-time health insights as both an incentive for participation and a tool for proactive disease management. These findings suggest that integrating biometric monitoring, symptom tracking, and patient-reported outcomes into clinical workflows during pragmatic trials could enhance overall disease management and patient satisfaction. This preference is significant in COPD research, where patients' clinical status can change quickly, and frequent acute care needs are a hallmark of the disease. This finding should be interpreted with caution: in many cases, participants are notified in advance that clinical data are observed but not communicated because the clinical implications of the data are unknown. Additionally, one must consider the potential for influencing study outcomes and confounding data by sharing study-based clinical information with patients in real-time. Investigators should consider a shared decision-making approach to managing study data, and prioritizing making interpretable data available, but also proactively sharing it with the relevant clinical teams so that it can be incorporated into clinical decision-making when applicable.

In addition to outcome selection, participants reported that study interventions designed to be accessible to patients living with COPD play a crucial role in engagement. Patients discussed the physical burden of participation, emphasizing that research should accommodate the mobility limitations and fatigue associated with COPD. Many preferred virtual or home-based participation options to reduce logistical barriers, a finding consistent with a larger body of research supporting remote and hybrid models of clinical trial engagement [33]. Additionally, sustained relationships with study staff were identified as a major factor in establishing and maintaining effective participant engagement. Participants recalled that personalized check-ins and ongoing communication fostered trust and motivation, reinforcing the importance of long-term participant-researcher relationships in study retention.

Finally, patients expressed a desire to be partners, rather than just participants, in research studies, moving beyond being passive study subjects to active contributors. They wanted to understand how their data was being used and to see and interpret the direct impact of their participation on scientific discovery and patient care. Many participants viewed research as an opportunity to contribute to the broader COPD community, expressing a sense of purpose and empowerment in advancing knowledge and improving care for themselves and future patients. These findings support the growing emphasis on patient-centered research approaches, which position participants as partners in the research process. Study teams might consider including patient co-investigators and advisory boards and conducting design studios during all phases of investigation.

The user profiles and FBM map developed during the study highlighted the heterogeneity in patient experiences with COPD and their varying priorities for, and ability to engage in, research, highlighting the need to actively involve a variety of patients in COPD research design to capture their diverse needs. However, despite the variation in perspectives, the profiles further underscored several central themes elicited in the interviews, emphasizing the need for personalized, patient-centered approaches in COPD research, ensuring that technologies and study designs align with individual capabilities, preferences, and long-term support needs.

## Limitations

This study has several limitations. The results of this qualitative project are intended to provide context to observational information and not establish causation [34]. Findings should not be interpreted as comprehensive or externally generalizable. Participants were predominantly white, digitally literate, and recruited from a single region, which may limit the applicability of findings to broader COPD populations, particularly those in non-urban or resource-limited settings [35,36]. Patients' occupations and socio-economic status were not tracked, and the study team did not perform subgroup analysis

by participant demographic features. Thus, their perspectives may not fully reflect the views of individuals who are hesitant or unwilling to participate in clinical research studies. The study response rate was modest; reasons for this could include that participants' contact information was incorrect or out of date, or that some participants were not comfortable participating in an interview or using teleconferencing software. These factors may have introduced unmeasured sampling bias into this study. The study was limited to English-speaking patients and may not reflect the values of all communities. Finally, other key informants, such as physicians, social workers, and other stakeholders who work with patients living with COPD, were not included in this study. Future studies should aim to deeply explore the perspectives of professional and lay key informants, as well as traditionally underrepresented groups and individuals with more severe COPD, and examine strategies for engaging those who are less likely to enroll in clinical studies [37]. It might also explore the perspectives of subgroups of patients to determine whether a certain research design strategy would help engage certain populations.

## Conclusions

This study emphasizes the need for patient-centered COPD research that prioritizes meaningful outcome measurement, minimizes participation burdens that preclude participation in research studies and clinical trials, and fosters ongoing engagement. Patients underlined the importance of measuring functional capacity, emotional well-being, and caregiver burden, ensuring that study data are not only actionable and clinically relevant but also proactively facilitating the translation of evidence into clinical adoption. As COPD research continues to evolve, active patient involvement in study design and execution will be essential to developing interventions that are both effective and widely implemented.

## Supporting information

**S1 File. Interview guide.** The final interview guide was used with participants.
(PDF)

**S1 Fig. Participant profiles.** Profiles of each participant were created to represent their perspective towards COPD-focused research, including key barriers and facilitators to engagement.
(PDF)

## Author contributions

**Conceptualization:** Laurel O'Connor, Anuska Ganesh Harne, Leah Dunkel, Peter Lindenauer, Bruce Miller, Christopher Mosher, Fernando Martinez, Apurv Soni.

**Data curation:** Laurel O'Connor, Leah Dunkel, Apurv Soni.

**Formal analysis:** Laurel O'Connor, Leah Dunkel.

**Funding acquisition:** Laurel O'Connor.

**Investigation:** Laurel O'Connor, Julia Ferranto, Anuska Ganesh Harne, Leah Dunkel.

**Methodology:** Laurel O'Connor, Julia Ferranto, Anuska Ganesh Harne, Leah Dunkel.

**Supervision:** Laurel O'Connor, Apurv Soni.

**Writing – original draft:** Laurel O'Connor.

**Writing – review & editing:** Julia Ferranto, Anuska Ganesh Harne, Leah Dunkel, Peter Lindenauer, Bruce Miller, Christopher Mosher, Fernando Martinez, Apurv Soni.

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
