## [Decision Letter · Decision Letter 0]

4 Jun 2025

Dear Dr. O'Connor,

Thank you for submitting your manuscript to PLOS ONE. After careful consideration, we feel that it has merit but does not fully meet PLOS ONE’s publication criteria as it currently stands. Therefore, we invite you to submit a revised version of the manuscript that addresses the points raised during the review process.

We look forward to receiving your revised manuscript.

Kind regards,

Natasha Shaukat

Academic Editor

PLOS ONE

Journal Requirements:

2. In the ethics statement in the Methods, you have specified that verbal consent was obtained. Please provide additional details regarding how this consent was documented and witnessed, and state whether this was approved by the IRB

3. We noted in your submission details that a portion of your manuscript may have been presented or published elsewhere. [A preliminary version of this manuscript was uploaded to the preprint service MedRXIV.] Please clarify whether this conference proceeding or publication was peer-reviewed and formally published. If this work was previously peer-reviewed and published, in the cover letter please provide the reason that this work does not constitute dual publication and should be included in the current manuscript.

4. Thank you for stating the following financial disclosure: [This project was supported by the National Center for Advancing Translational Sciences, National Institutes of Health, through Grant KL2TR00145]. 

5. We notice that your supplementary file are included in the manuscript file. Please remove them and upload them with the file type 'Supporting Information'. Please ensure that each Supporting Information file has a legend listed in the manuscript after the references list.

Additional Editor Comments:

Please see reviewers comments at the end.

Reviewers' comments:

Reviewer's Responses to Questions

**Comments to the Author**

1. Is the manuscript technically sound, and do the data support the conclusions?

Reviewer #1: Yes

Reviewer #2: Yes

2. Has the statistical analysis been performed appropriately and rigorously?

Reviewer #1: Yes

Reviewer #2: I Don't Know

3. Have the authors made all data underlying the findings in their manuscript fully available?

Reviewer #1: Yes

Reviewer #2: Yes

4. Is the manuscript presented in an intelligible fashion and written in standard English?

Reviewer #1: Yes

Reviewer #2: Yes

Reviewer #1: Congratulations to the authors on their well-conducted and insightful research.

The authors used the Theoretical Framework of Acceptability (TFA) and the Fogg Behavior Model (FBM) to guide their data collection and analysis, which was apt. The study brings important insights into what matters most to people living with COPD.

There are several notable strengths. The methodological rigor is further demonstrated through high inter-coder reliability (Krippendorff’s alpha of 0.86) and evidence of thematic saturation. The qualitative data are rich, and the authors clearly present their findings, supported by explanatory participant quotes.

There are, however, a few areas where the manuscript could be improved:

• The FBM mapping is not presented visually. Maybe if the authors couldinclude a figure or some form that illustrates this would help readers better understand how the model was applied.

• the manuscript does not provide sufficient detail about who the caregivers are—for example, their relationship to the patient, or the intensity of their caregiving roles. A table or brief summary of this information would be useful.

• The study is based in a single region and includes a sample that appears to be largely White and digitally literate. The limitations section should more clearly acknowledge how this may affect the generalisability of the findings to other populations, particularly those from more diverse or underserved backgrounds..

• It would be helpful if the authors briefly described in the Methods section how the interview guide evolved after piloting.

• The manuscript mentions participant personas as separate illustrations in the supplementary materials. It would be good to clarify if these were developed solely for internal understanding or were intended for dissemination or further use.

• The authors have done an excellent job of centering patient and caregiver voices in their exploration of COPD clinical research. With a few minor revisions to improve clarity and transparency, I believe this manuscript will make a valuable addition to the literature.

Reviewer #2: This research explored what matters most to patients with COPD in order to better align COPD research with their real-life needs and experiences. This is a timely and interesting study. I have a few comments:

1. Did you consider the occupation of the participants? Or income? It should have a significant impact on understanding the patient’s needs/perception.

2. Can you please comment on the response rate?

3. Prior understanding/research of this topic is not well discussed.

4. In the discussion, it will be impactful to have discussions of possible barriers to addressing the patient’s needs and how to address these issues in real life.

5. Did you consider the subgroup-level analysis? (for example: by gender)

6. Did you consider key informant interviews? It can guide future implementation of research findings.

Relevant articles:

Sigurgeirsdottir, J., Halldorsdottir, S., Arnardottir, R. H., Gudmundsson, G., & Bjornsson, E. H. (2019). COPD patients' experiences, self-reported needs, and needs-driven strategies to cope with self-management. International journal of chronic obstructive pulmonary disease, 14, 1033–1043. https://doi.org/10.2147/COPD.S201068

Poletti, V., Bresciani, G., Banfi, P., & Volpato, E. (2024). Exploring perceptions and expectations of COPD patients: A grounded theory approach for personalized therapeutic interventions. Chronic respiratory disease, 21, 14799731241268262. https://doi.org/10.1177/14799731241268262

**Do you want your identity to be public for this peer review?** For information about this choice, including consent withdrawal, please see our Privacy Policy

Reviewer #1: **Yes: ** Dr. Adrija Roy

Reviewer #2: No

---

## [Author Response · Author response to Decision Letter 1]

9 Jun 2025

June 4, 2025

Dear Dr. Shaukat,

We sincerely appreciate your time and expertise, and that of our reviewers. We are grateful for their thoughtful, critical reviews and believe that responding to their suggestions has significantly strengthened the manuscript.

Below, please find a line-by-line response to the reviewers’ comments and the revisions made in response to their suggestions. For clarity, the authors’ responses and actions taken have been bolded. We hope that you find the manuscript substantially improved and suitable for publication.

Author Response: The authors understand the importance of formatting the manuscript in the style of PLOS ONE.

Action Taken: The manuscript has been carefully revised to comply with all PLOS ONE’s formatting requirements.

2. In the ethics statement in the Methods, you have specified that verbal consent was obtained. Please provide additional details regarding how this consent was documented and witnessed, and state whether this was approved by the IRB

Author Response: The authors agree that further clarification of our consent process is needed. Because the study is considered less than minimal risk by the authorizing IRB (entailing only activities that would not require written consent outside of the research context), verbal consent was allowed. Per our institutional protocol, a written Fact Sheet was provided to all participants to read and review before deciding to consent and begin study activities. The participant was then allowed to provide verbal consent witnessed by at least one additional study team member, and the interview proceeded. The verbal consent was documented on the cover sheet of each participant’s interview notes. This process was authorized by the supervising IRB of record.

Action Taken: The methods section of the manuscript was revised to note that all participants were provided with a written Fact Sheet at the time of recruitment and provided fully informed verbal consent to a member of the study team. A second member of the study team witnessed the consent before the start of the interview. The consent process and participant-facing Fact Sheet were approved under the authorizing IRB and complied with all relevant ethical regulations, including the Declaration of Helsinki.

3. We noted in your submission details that a portion of your manuscript may have been presented or published elsewhere. [A preliminary version of this manuscript was uploaded to the preprint service MedRXIV.] Please clarify whether this conference proceeding or publication was peer-reviewed and formally published. If this work was previously peer-reviewed and published, in the cover letter please provide the reason that this work does not constitute dual publication and should be included in the current manuscript.

Author Response: The authors note that the manuscript has been uploaded to a preprint server (MedRVIX), but it has not been formally peer-reviewed, formally published, or presented at an academic conference and therefore this submission does not constitute dual publication.

Action Taken: The submission details and cover letter have been updated to note that A preliminary version of this manuscript was submitted to the preprint service MedRVIX, but it has not been formally peer-reviewed, published, or presented at an academic conference.

4. Thank you for stating the following financial disclosure: [This project was supported by the National Center for Advancing Translational Sciences, National Institutes of Health, through Grant KL2TR00145].

Please state what role the funders took in the study. If the funders had no role, please state: ""The funders had no role in study design, data collection and analysis, decision to publish, or preparation of the manuscript." If this statement is not correct you must amend it as needed. Please include this amended Role of Funder statement in your cover letter; we will change the online submission form on your behalf.

Author Response: The authors agree with the editor that it is important to clarify the role of funders in any research project.

Action Taken: The cover letter has been revised to note that the project was supported by the National Center for Advancing Translational Sciences, National Institutes of Health, through Grants KL2TR001454 and UL1-TR001453. The funders had no role in study design, data collection and analysis, decision to publish, or preparation of the manuscript.

5. We notice that your supplementary file are included in the manuscript file. Please remove them and upload them with the file type 'Supporting Information'. Please ensure that each Supporting Information file has a legend listed in the manuscript after the references list.

Author Response: The authors regret this error

Action Taken: The supplemental file has been reuploaded in the appropriate section in the submission portal.

Author Response: The citations have been thoroughly reviewed and are correct and complete. In response to one of the reviewers’ feedback, we have added two additional references, which are discussed below.

Action Taken: Two additional references were added at the suggestion of a reviewer. No references were removed.

Reviewers' comments:

Comments to the Author

1. Is the manuscript technically sound, and do the data support the conclusions?

Reviewer #1: Yes

Reviewer #2: Yes

Author Response: The authors are glad that the reviewers found the manuscript technically sound.

Action Taken: No action taken

2. Has the statistical analysis been performed appropriately and rigorously?

Reviewer #1: Yes

Reviewer #2: I Don't Know

Author Response: There was no statistical analysis performed for this qualitative study.

Action Taken: No action taken

3. Have the authors made all data underlying the findings in their manuscript fully available?

Reviewer #1: Yes

Reviewer #2: Yes

Author Response: The authors are glad that the reviewers believe that the findings were fully available

Action Taken: No action taken

4. Is the manuscript presented in an intelligible fashion and written in standard English?

Reviewer #1: Yes

Reviewer #2: Yes

Author Response: The authors are glad that the reviewers feel that the manuscript was written in an intelligible fashion

Action Taken: No action taken

5. Review Comments to the Author

Reviewer #1: Congratulations to the authors on their well-conducted and insightful research.

The authors used the Theoretical Framework of Acceptability (TFA) and the Fogg Behavior Model (FBM) to guide their data collection and analysis, which was apt. The study brings important insights into what matters most to people living with COPD. There are several notable strengths. The methodological rigor is further demonstrated through high inter-coder reliability (Krippendorff’s alpha of 0.86) and evidence of thematic saturation. The qualitative data are rich, and the authors clearly present their findings, supported by explanatory participant quotes.

Author Response: The authors are glad that the reviewer found the manuscript rigorous and insightful

Action Taken: No action taken

There are, however, a few areas where the manuscript could be improved:

• The FBM mapping is not presented visually. Maybe if the authors could include a figure or some form that illustrates this would help readers better understand how the model was applied.

Author Response: The authors agree that further explanation of how the FBM model was applied, and how the results were interpreted, would be helpful in the manuscript

Action Taken: Additional interpretation of the application of the FBM was added to the discussion. Additionally, we have added “Figure 1”, depicting the mapping of each participant onto an FBM graph to provide readers with an illustration of the results.

• the manuscript does not provide sufficient detail about who the caregivers are—for example, their relationship to the patient, or the intensity of their caregiving roles. A table or brief summary of this information would be useful.

Author Response: The authors agree that additional information about the caretakers included in the study would be helpful.

Action Taken: A brief summary of the two caretaker participants’ relationship and role with respect to a patient living with COPD was added to the beginning of the results section.

• The study is based in a single region and includes a sample that appears to be largely White and digitally literate. The limitations section should more clearly acknowledge how this may affect the generalisability of the findings to other populations, particularly those from more diverse or underserved backgrounds..

Author Response: The authors agree with the reviewer that the relatively homogenous sample represented in this study may limit generalizability to the larger population of patients living with COPD.

Action Taken: Additional language was added to the limitations, highlighting that the small, relatively homogenous sample represented in this study may not be generalizable to the entire population of patients living with COPD and that further study specifically evaluating the perspectives of populations underrepresented in research is warranted.

• It would be helpful if the authors briefly described in the Methods section how the interview guide evolved after piloting.

Author Response: The authors agree with the reviewer that a description of the interview guide modification process would help readers understand the methodological approach to this study.

Action Taken: A description of how and when the interview guide was modified was added to the methods section of the paper.

• The manuscript mentions participant personas as separate illustrations in the supplementary materials. It would be good to clarify if these were developed solely for internal understanding or were intended for dissemination or further use.

Author Response: The reviewer asks a good question. The personas developed during the study were for internal use only to spark discussion and guide future patient-centered study design. However, we also felt that they would be of interest to readers and therefore were shared as a supplemental file.

Action Taken: The methods were revised to specifically note that the personas were developed to internally guide future study design.

The authors have done an excellent job of centering patient and caregiver voices in their exploration of COPD clinical research. With a few minor revisions to improve clarity and transparency, I believe this manuscript will make a valuable addition to the literature.

Author Response: The authors are glad that the reviewer feels that this manuscript will make a meaningful contribution to the literature

Action Taken: No action taken

Reviewer #2: This research explored what matters most to patients with COPD in order to better align COPD research with their real-life needs and experiences. This is a timely and interesting study. I have a few comments:

Author Response: The authors are gratified that the reviewer found the manuscript relevant and interesting.

Action Taken: No action taken

1. Did you consider the occupation of the participants? Or income? It should have a significant impact on understanding the patient’s needs/perception.

Author Response: We did not track the income or occupation of the participants. The authors agree with the reviewer that these factors impact patients’ perspectives and that not reporting on them may limit generalizability to the larger population of patients living with COPD.

Action Taken: Additional language was added to the limitations, highlighting that the small, relatively homogenous sample represented in this study, as well as the fact that occupation and SES was not reported on, may result in a sample whose perspective is not generalizable to the entire population of patients living with COPD and that further study specifically evaluating the perspectives of populations underrepresented in research is warranted.

2. Can you please comment on the response rate?

Author Response: The authors agree with the reviewer that a comment on the response rate would be appropriate in the discussion of this paper. Some participants’ contact information may have been incorrect or out of date, some participants may not be comfortable participating in an interview or using teleconferencing software. We should acknowledge that the response rate

Action Taken: The authors added a brief discussion of the modest participant response rate and the possibility of resultant sampling bias to the limitations of the paper.

3. Prior understanding/research of this topic is not well discussed.

Author Response: The authors agree with the reviewer that a discussion of prior research exploring this topic was modest in the discussion, and the paper would benefit from additional examination. We are grateful for the suggested citations.

Action Taken: The authors added a more robust assessment of the prior literature on the topic of patient priorities in COPD, and compared it to our findings.

4. In the discussion, it will be impactful to have discussions of possible barriers to addressing the patient’s needs and how to address these issues in real life.

Author Response: The authors agree that discussing the barriers to patient preferences in more detail would be interesting and important for readers. We did discuss some of the barriers and solutions to real-time data sharing, but agree that we should expand this approach to the study’s other major findings.

Action Taken: The authors added exploration of the barriers and solutions to addressing patient preferences in COPD research to the project’s discussion.

5. Did you consider the subgroup-level analysis? (for example: by gender)

Author Response: The authors agree with the reviewer that a subgroup analysis of patient perspectives by discrete demographic features would be relevant and interesting. However, as this was a qualitative study, our goal was to explore the depth and range of participant experiences rather than to compare groups. Given the sample size and the exploratory nature of the study, we prioritized thematic saturation and var

---

## [Decision Letter · Decision Letter 1]

15 Aug 2025

“A Good Day Is Just Being Able to Breathe": Aligning COPD Research with Patient Needs, a Qualitative Study

PONE-D-25-18562R1

Dear Dr. Connor ,

We’re pleased to inform you that your manuscript has been judged scientifically suitable for publication and will be formally accepted for publication once it meets all outstanding technical requirements.

Kind regards,

Natasha Shaukat

Academic Editor

PLOS ONE

Reviewers' comments:

Reviewer's Responses to Questions

**Comments to the Author**

Reviewer #1: All comments have been addressed

2. Is the manuscript technically sound, and do the data support the conclusions?

Reviewer #1: Yes

3. Has the statistical analysis been performed appropriately and rigorously?

Reviewer #1: Yes

4. Have the authors made all data underlying the findings in their manuscript fully available?

Reviewer #1: Yes

5. Is the manuscript presented in an intelligible fashion and written in standard English?

Reviewer #1: Yes

Reviewer #1: Thank you for addressing all the comments. The manuscript is now okay for publication from my end. Accepted.

**Do you want your identity to be public for this peer review?** For information about this choice, including consent withdrawal, please see our Privacy Policy

Reviewer #1: **Yes: ** Adrija Roy

---

## [Editor Report · Acceptance letter]

PONE-D-25-18562R1

PLOS ONE

Dear Dr. O'Connor,

I'm pleased to inform you that your manuscript has been deemed suitable for publication in PLOS ONE. Congratulations! Your manuscript is now being handed over to our production team.

Kind regards,

on behalf of

Dr. Natasha Shaukat

Academic Editor

PLOS ONE